# ENABLING FINE-TUNING OF DIRECT FEEDBACK ALIGNMENT VIA FEEDBACK-WEIGHT MATCHING

## ABSTRACT

In this paper, we introduce feedback-weight matching, a new method that facilitates reliable fine-tuning of fully connected neural networks using Direct Feedback Alignment (DFA). Although DFA has demonstrated potential by enabling efficient and parallel updates of weight parameters through direct propagation of the network's output error, its usage has been primarily restricted to training networks from scratch. We provide the first analysis showing that existing standard DFA struggles to fine-tune networks pre-trained via back-propagation. Through an analysis of weight alignment (WA) and gradient alignment (GA), we show that the proposed feedback-weight matching enhances DFA's ability and stability in fine-tuning pre-trained networks, providing insights into DFA's behavior and characteristics when applied to fine-tuning. In addition, we find that feedback-weight matching, when combined with weight decay, not only mitigates over-fitting but also further reduces the network output error, leading to improved learning performance during DFA-based fine-tuning. Our experimental results show that, for the first time, feedback-weight matching enables reliable and superior fine-tuning across various fine-tuning tasks compared to existing standard DFA, e.g., achieving 7.97% accuracy improvement on image classification tasks (i.e., 82.67% vs. 74.70%) and 0.66 higher correlation score on NLP tasks (i.e., 0.76 vs. 0.10). The code implementation is available at an anonymous GitHub repository[1].

## 1 INTRODUCTION

Recently, a new training mechanism called Direct Feedback Alignment (DFA) (Nøkland, 2016) has been proposed for deep neural networks to alleviate the weight transport problem (Grossberg, 1987; Crick, 1989). Based on the concept of Feedback Alignment (FA) (Lillicrap et al., 2016), DFA passes the error of the output layer directly to each layer of the network to update the weight parameters without back-propagation (Rumelhart et al., 1986). By using random feedback matrices, the weight gradient of each layer is independently approximated from the directly passed error, enabling efficient training of networks through the parallel update of multiple layers. This contrasts with back-propagation that propagates the network error sequentially from the last to the first layer.

Although Direct Feedback Alignment (DFA) (Nøkland, 2016) has shown its potential in training primarily for fully connected networks (Garg & Vempala, 2022; Launay et al., 2020), its application to fine-tuning (Devlin et al., 2018), i.e., adapting a pre-trained network to a new task, has been less studied until today despite its practical usefulness. In fact, it has been known that fine-tuning networks with DFA is challenging (Chu & Bacho, 2024); the performance of networks fine-tuned with DFA is generally unreliable compared to that of those fine-tuned with back-propagation (Rumelhart et al., 1986). Given that fine-tuning has become one of the practical and also effective ways of re-utilizing pre-trained networks for various downstream tasks (Church et al., 2021), investigating how DFA can be applied to the fine-tuning mechanism both theoretically and empirically is necessary.

Enabling fine-tuning with Direct Feedback Alignment (DFA) (Nøkland, 2016) can not only broaden DFA's usability but also introduce an alternative approach to current back-propagation-based fine-tuning (Rumelhart et al., 1986; Church et al., 2021). Currently, DFA has not yet been established as a reliable stand-alone training method that can provide comparable performance to back-propagation (Launay et al., 2019; Crafton et al., 2019). Thus, taking a wide range of well-pre-trained

---

[1] https://anonymous.4open.science/r/Feedback-Weight-Matching-C7F0

models, such as Transformer-based foundation models (Kenton & Toutanova, 2019), as the starting point would be a practical strategy that can complement DFA's unstable and limited learning capabilities. Additionally, by incorporating DFA's unique advantages, such as being back-propagation-free and enabling parallel training, into the widely used fine-tuning scheme, we can explore new possibilities for re-utilizing pre-trained models in a more agile, efficient, and biologically plausible manner, in contrast to conventional back-propagation, which requires significantly more resources and time.

In this paper, we introduce a DFA-based fine-tuning method, which investigates the feasibility of Direct Feedback Alignment (DFA) (Nøkland, 2016) for fine-tuning deep neural networks, with the aim of extending the scope of DFA to embrace various pre-trained networks. We first analyze the reasons why the existing standard DFA, which updates the pre-trained weights using random feedback matrices, does not perform well in fine-tuning. This analysis is based on the weight alignment (WA) and gradient alignment (GA) (Refinetti et al., 2021), which are two measures proposed to estimate the state and learning performance of DFA. From this analysis, we propose the feedback-weight matching, which first reconstructs the feedback matrices by decomposing the pre-trained weights and then re-initializes the weights based on the reconstructed feedback matrices before starting fine-tuning. Additionally, we prove that applying weight decay (Krogh & Hertz, 1991) on top of feedback-weight matching considerably improves and stabilizes the fine-tuning performance of DFA, beyond the general regularization effect on weight parameters. Together with the simple yet effective feedback-weight matching, weight decay acts as a key facilitator for fine-tuning fully connected networks with DFA. To the best of our knowledge, this work is the first attempt to explore the possibility of applying DFA to fine-tuning of fully connected networks via an in-depth study.

The experiments provide evaluation results consistent with our theoretical analysis; applying feedback-weight matching enables more effective and reliable fine-tuning of fully connected networks with DFA over various fine-tuning tasks, when compared to the existing standard DFA (Nøkland, 2016) that does not apply the proposed feedback-weight matching. For instance, the image classification accuracy of fully connected networks fine-tuned with feedback-weight matching reaches 82.67%, while that of standard DFA remains 74.70%. Also, it successfully fine-tunes Transformer models (BERT) (Devlin et al., 2018) on NLP tasks, e.g., achieving 0.76 correlation score, while the standard DFA barely conducts fine-tuning at all, i.e., achieving mere 0.10 correlation score. The results demonstrate the potential for extending DFA towards the widely used pre-training and fine-tuning strategy, moving beyond its limited usage in from-scratch training.

## 2 BACKGROUND AND RELATED WORK

**DFA**. It is common to train a neural network using the back-propagation algorithm (Rumelhart et al., 1986). Given a fully connected network, we denote $W_l$ as the weight of $l$-th layer of the network, $\mathcal{L}(\hat{y}, y)$ as the loss function, where $\hat{y}$ is the ground-truth output, and $y$ is the network output, and $h_l = g(a_l)$ as the output of the $l$-th layer, where $g(\cdot)$ is activation function, and $a_l = W_l h_{l-1}$. To update the weight with the gradient descent algorithm (Ruder, 2016), the gradient of the loss $\mathcal{L}$ w.r.t. the weight $W_l$ is obtained using back-propagation (BP) as:

$$\delta W_l^{BP} = -\frac{\partial \mathcal{L}}{\partial W_l} = -\left[(W_{l+1}^\top \delta a_{l+1}) \odot g'(a_l)\right] h_{l-1}^\top, \ \delta a_l = \frac{\partial \mathcal{L}}{\partial a_l} \quad (1)$$

where $\odot$ is the Hadamard product. However, back-propagation poses some challenges, specifically the weight transport (Grossberg, 1987; Crick, 1989) and backward locking problems (Lillicrap et al., 2020; Launay et al., 2019). Direct Feedback Alignment (DFA) (Nøkland, 2016) addresses the weight transport problem by employing random feedback and mitigates the backward locking problem by delivering the network's output error signal to each layer independently. Specifically, 1) the global error vector $e = \hat{y} - y$ is transmitted to each layer, and 2) the weight $W_{l+1}$ at the $l$-th layer of the network is replaced with a random feedback matrix $F_l$, leading to the following weight gradient:

$$\delta W_l^{DFA} = -\frac{\partial \mathcal{L}}{\partial W_l} = -\left[(F_l e) \odot g'(a_l)\right] h_{l-1}^\top - \lambda^t W_l \quad (2)$$

where $\lambda^t$ is the weight-decay hyperparameter at the step $t$. Equation (2) eliminates the necessity of sequential layer-wise gradient computations required by back-propagation (Rumelhart et al., 1986).

**GA and WA**. To better elucidate the dynamics of DFA (Nøkland, 2016), the concept of gradient alignment (GA) is introduced (Lillicrap et al., 2016). GA quantitatively assesses the similarity be-

tween the weight gradients obtained through DFA and those derived via back-propagation (Rumelhart et al., 1986). This is achieved by comparing the weight updates generated from the identically initialized weights by both methods. It has been hypothesized that a stronger (higher) GA corresponds to enhanced learning performance in DFA. In addition, the concept of weight alignment (WA) (Refinetti et al., 2021) has been introduced to evaluate the relationship between the weight and the feedback matrix in DFA, suggesting that strong WA is associated with strong GA. Although GA and WA have been instrumental in analyzing the learning efficacy of DFA, prior research has not explored their utility in the context of fine-tuning. In contrast, this paper pioneers the application of GA and WA concepts to systematically investigate the fine-tuning process in DFA.

**Applicability to Transformers and CNNs**. Some studies (Launay et al., 2020) explore the applicability of DFA (Nøkland, 2016) to various fully connected networks, including neural radiance fields (NeRF) (Mildenhall et al., 2021; Sitzmann et al., 2019), recommender systems (Guo et al., 2017), and NLP (Vaswani, 2017; Merity et al., 2016). While they show that DFA can train a wide range of deep architectures, they also reveal a significant performance gap between DFA and back-propagation (Rumelhart et al., 1986), particularly in Transformer models (Vaswani, 2017). When applied to models not based on fully connected networks, such as CNNs, the performance gap between DFA and back-propagation is even more pronounced. For instance, VGG-16 (Simonyan & Zisserman, 2014) on CIFAR-100 (Krizhevsky et al., 2009) trained with DFA achieves 1% top-1 accuracy (Launay et al., 2019), while back-propagation achieves 60%. Similarly, in ImageNet (Deng et al., 2009), it is 6.2% vs. 53% (Crafton et al., 2019). Given that applying DFA to from-scratch training scenarios 1) consistently underperforms relative to back-propagation, 2) takes a much longer training time than fine-tuning, and 3) is limited to a narrower range of architectures, we argue that utilizing DFA for fine-tuning would be a more effective, efficient, practical, and expedient approach. Thus, in this study, we investigate and analyze the potential of employing DFA in fine-tuning, which is conducive to the widely-used pre-train-and-fine-tune strategy (Devlin et al., 2018).

**Applying DFA to Back-Propagation Weights**. As described above, in CNNs, DFA encounters challenges in effectively learning the necessary spatial information (Crafton et al., 2019). Similarly, in fully connected networks, DFA is known to produce feature representation clusters that deviate from those learned via back-propagation (Nøkland, 2016). Moreover, although stable training can be achieved when transitioning from weights learned through DFA to back-propagation, the reverse is not true; switching from back-propagation to DFA results in unstable training, and DFA fails to fully recover its performance even after large training epochs (Chu & Bacho, 2024). These imply the inherent difficulties in fine-tuning with DFA using weights pre-trained with back-propagation.

**DFA with Weight Decay**. In the study by Song et al. (2021), it is analyzed that weight decay (Krogh & Hertz, 1991) can reduce the output error in fully connected networks when used with Feedback Alignment (FA) (Lillicrap et al., 2016). Nevertheless, the analysis predominantly focuses on the training of networks from scratch using FA, rather than on the fine-tuning process with DFA. This work, for the first time, examines the impact of weight decay in the context of fine-tuning with DFA. Our findings indicate that weight decay can be beneficial in fine-tuning with DFA, as it reduces network output error and over-fitting, thereby enhancing overall learning performance.

## 3 FEEDBACK-WEIGHT MATCHING

We first discuss why the existing standard DFA (Nøkland, 2016) does not behave stably in fine-tuning, based on weight alignment (WA) and gradient alignment (GA) (Refinetti et al., 2021). Then, we introduce feedback-weight matching, which enables effective and reliable fine-tuning of DFA.

### 3.1 WHY DOES DFA PERFORM UNRELIABLY IN FINE-TUNING?

**Definition 3.1.** *(Weak Weight Alignment)* Given a $L$-layer fully connected linear network updated (trained) with DFA (Nøkland, 2016), the weight of the $l$-th layer at the $t$-th training step, which is denoted as $\boldsymbol{W}_{1 \leq 1 \leq L}^t$, becomes (Refinetti et al., 2021) as follows:

$$\boldsymbol{W}_1^t = \boldsymbol{F}_1 \boldsymbol{A}_1^t, \ \boldsymbol{W}_{1<l<L}^t = \boldsymbol{F}_l \boldsymbol{A}_l^t \boldsymbol{F}_{l-1}^\top, \ \text{and} \ \boldsymbol{W}_L^t = \boldsymbol{A}_L^t \boldsymbol{F}_{L-1}^\top,$$

$$\text{where} \ \boldsymbol{A}_1^t = -\eta \sum\nolimits_{t'=0}^{t-1} \boldsymbol{e}^{t'} (\boldsymbol{x}^{t'})^\top, \ \text{and} \ \boldsymbol{A}_{l \geq 2}^t = \eta^2 \sum_{t'=0}^{t-1} \sum_{t''=0}^{t'-1} (\boldsymbol{B}_l^{t'} \boldsymbol{x}^{t'}) \cdot (\boldsymbol{B}_l^{t''} \boldsymbol{x}^{t''}) \boldsymbol{e}^{t'} (\boldsymbol{e}^{t''})^\top \quad (3)$$

Here, $F_l$ is the feedback matrix of the $l$-th layer, $A_1^t$ and $A_{l \geq 2}^t$ are the alignment matrices, and $B_l = A_{l-2} \cdots A_0 \in \mathbb{R}^{n_L \times n_L}$ is defined recursively using the feedback matrices only, with $A_0 = I$ (Refinetti et al., 2021). Equation (3) is referred to as *weak weight alignment (WA)* (Refinetti et al., 2021), representing the state where no particular relationship exists between $W_{1 < l < L}^t$ and $F_l F_{l-1}^\top$ and between $W_L^t$ and $F_{L-1}^\top$. At the early stage of DFA training, weak WA is naturally induced since $A_{l \geq 2}^t$ in Equation (3) starts with arbitrary values. However, as the training proceeds, $A_{l \geq 2}^t$ becomes proportional to the identity matrix (Refinetti et al., 2021), i.e., $A_{l \geq 2}^t \propto I$, leading to another state called *strong weight alignment (WA)*, which is defined as follows.

**Definition 3.2.** *(Strong Weight Alignment)* If $A_{l \geq 2}^t \propto I$, Equation (3) becomes the state called *strong weight alignment (WA)*, which is defined as follows.

$$W_{1 < l < L}^t \propto F_l F_{l-1}^\top, \ W_L^t \propto F_{L-1}^\top \tag{4}$$

It is known that the strong WA in Equation (4), given $F_l^\top F_l \equiv I$, implies *strong gradient alignment (GA)* (Refinetti et al., 2021) defined in Equation (9), causing the gradient direction of the DFA weight, $W_{1 < l \leq L}^t$, aligned to that of back-propagation (Rumelhart et al., 1986). Hence, strong WA leads the learning trajectory of DFA to be comparable to that of back-propagation with strong GA.

However, if the pre-trained weights are fine-tuned via existing standard DFA using arbitrary random feedback matrix $F_l$, it becomes difficult to achieve strong WA in Equation (4), as shown below, likely to result in sub-optimal fine-tuning performance by inducing weak GA from weak WA.

**Proposition 3.3.** *If the pre-trained weight, denoted as $W_l^0$, is updated using DFA with arbitrary random feedback matrices $F_l$, the strong WA condition in Equation (4) is unlikely to be satisfied.*

$$W_{1 < l < L}^t \not\propto F_l F_{l-1}^\top, \ W_L^t \not\propto F_{L-1}^\top \tag{5}$$

where $W_l^t$ denotes the weight after $t$ steps of training, starting from the pre-trained weight $W_l^0$. Equation (5) shows that the weight trained from the backpropagation pre-trained weight does not satisfy the strong WA condition. The proof is detailed in Appendix A.

### 3.2 INDUCING STRONG WEIGHT ALIGNMENT

To enable fine-tuning with DFA by deriving strong GA from strong WA defined in Equation (4), we propose the *feedback-weight matching* method, which induces both strong WA and GA as follows.

**Definition 3.4.** *(Feedback Matching)* From the pre-trained weight $W_l^0$, we set the feedback matrix $\bar{F}_l$ such that:

$$\bar{F}_l \bar{F}_{l-1}^\top \equiv W_{1 < l < L}^0 \text{ and } \bar{F}_{L-1}^\top \equiv W_L^0. \tag{6}$$

Equation (6) requires us to decompose the pre-trained weight $W_{1 < l < L}^0$ into $\bar{F}_l$ and $\bar{F}_{l-1}^\top$. It can be achieved either through traditional decomposition methods, such as SVD (Singular Value Decomposition) (Klema & Laub, 1980), or alternatively, by optimizing Equation (23) in Appendix B.

Once the feedback matrix $\bar{F}_l$ is reconstructed as Equation (6), we proceed to the *weight matching* process to induce strong WA, as described below.

**Definition 3.5.** *(Weight Matching)* Given the reconstructed $\bar{F}_l$ derived by feedback matching (Equation (6)), we re-initialize the pre-trained weight $W_l^0$ into $\bar{W}_l^0$ so that it matches $\bar{F}_l$ such that:

$$\bar{W}_{1 < l < L}^0 \equiv \bar{F}_l \bar{F}_{l-1}^\top \text{ and } \bar{W}_L^0 \equiv \bar{F}_{L-1}^\top. \tag{7}$$

The following shows that Equation (6) and (7) lead to strong WA condition in Equation (4).

**Proposition 3.6.** *If the re-initialized weight $\bar{W}_l^0$ in Equation (7) is updated using DFA with the feedback matrix $\bar{F}_l$ derived by Equation (6), the strong WA condition in Equation (4) is induced.*

$$\bar{W}_{1 < l < L}^t \propto \bar{F}_l \bar{F}_{l-1}^\top, \ \bar{W}_L^t \propto \bar{F}_{L-1}^\top \tag{8}$$

with $\bar{W}_l^t$ is the weight at step $t$, initialized from $\bar{W}_l^0$. Equation (8) indicates that the weight updated from the re-initialized weight satisfies the strong WA condition, the proof is detailed in Appendix A.

Subsequently, strong WA, achieved through Equation (6) and Equation (7), leads to strong GA (Refinetti et al., 2021). By matching the feedback matrix to the pre-trained weights, as in Equation (6), it becomes possible to preserve the knowledge embedded in the pre-trained weights. Additionally, by re-initializing the pre-trained weights from the matched feedback matrices, as in Equation (7), it becomes possible to facilitate the attainment of strong WA through DFA in fine-tuning.

### 3.3 INDUCING STRONG GRADIENT ALIGNMENT

While the previous section (Section 3.2) shows that the proposed feedback-weight matching in Equation (6) and (7) promotes strong weight alignment (WA), naturally leading to strong gradient alignment (GA), we now show that feedback-weight matching also directly induces strong GA.

**Definition 3.7.** *(Gradient Alignment)* The gradient alignment (GA) is defined as the cosine similarity between the weight gradient obtained using DFA (Nøkland, 2016), denoted $\boldsymbol{G}_{DFA}$, and the weight gradient of back-propagation (Rumelhart et al., 1986), denoted $\boldsymbol{G}_{BP}$, which is given by:

$$\cos \angle(\boldsymbol{G}_{DFA}, \boldsymbol{G}_{BP}) = \boldsymbol{G}_{DFA} \cdot \boldsymbol{G}_{BP} / \|\boldsymbol{G}_{DFA}\|\|\boldsymbol{G}_{BP}\|. \tag{9}$$

We show that feedback-weight matching, i.e., Equation (6) and (7), also directly induce strong GA when fine-tuning the first layer of the two-layer fully connected linear network, as follows.

**Proposition 3.8.** *Feedback-weight matching given in Equation (6) and (7) induces strong GA, i.e., a higher GA, in the first layer of a fully connected linear network.*

$$\cos_{FWM} \angle(\boldsymbol{F}_1, \boldsymbol{W}_2^t) \geq \cos_{DFA} \angle(\boldsymbol{F}_1, \boldsymbol{W}_2^t) \tag{10}$$

$\cos_{FWM} \angle(\boldsymbol{F}_1, \boldsymbol{W}_2^t)$ refers to GA in the first layer using feedback-weight matching, while $\cos_{DFA} \angle(\boldsymbol{F}_1, \boldsymbol{W}_2^t)$ refers to GA in the first layer without feedback-weight matching. Equation (10) shows the GA when feedback-weight matching is used and when it is not. The proof is detailed in Appendix A. Based on Proposition 3.8, which shows feedback-weight matching induces stronger GA, we provide the following conjecture, which generalizes it to an arbitrary $L$-layer fully connected linear network.

**Conjecture 3.9.** *It is conjectured that Equation (6) and (7) induce strong gradient alignment (GA), i.e., a higher GA, for all $1 \leq l \leq L$ layers in a fully connected linear network, where $L > 2$.*

## 4 WEIGHT DECAY

Similar to conventional training using back-propagation (Nøkland, 2016), weight decay (Krogh & Hertz, 1991) has been shown to mitigate over-fitting of DFA, though its effect in fine-tuning has not been studied. We discuss how the proposed feedback-weight matching helps weight decay to reduce the network error (i.e., improving learning performance) during fine-tuning when applied to DFA.

**Lemma 4.1.** *Given the re-initialized weight $\bar{\boldsymbol{W}}_{1<l\leq L}^0$ in Equation (7) and the pre-trained weight $\boldsymbol{W}_{1<l\leq L}^0$, the following terms, $r_{1<l<L}$ and $r_L$, are non-negative with high probability.*

$$r_{1<l<L} = \|\boldsymbol{W}_l^t - \boldsymbol{W}_l^0\| - \|\boldsymbol{W}_l^t - \bar{\boldsymbol{W}}_l^0\| = \|\bar{\boldsymbol{F}}_l \bar{\boldsymbol{F}}_{l-1}^\top - \boldsymbol{W}_l^0\| - |c_l^t - 1|\|\bar{\boldsymbol{F}}_l \bar{\boldsymbol{F}}_{l-1}^\top\| \geq 0 \tag{11}$$

$$r_L = \|\boldsymbol{W}_L^t - \boldsymbol{W}_L^0\| - |\boldsymbol{W}_L^t - \bar{\boldsymbol{W}}_L^0\| = \|\bar{\boldsymbol{F}}_{L-1}^\top - \boldsymbol{W}_L^0\| - |c_L^t - 1|\|\bar{\boldsymbol{F}}_{L-1}^\top\| \geq 0, \tag{12}$$

*implying that $\|\boldsymbol{W}_l^t - \boldsymbol{W}_l^0\| \geq |\boldsymbol{W}_l^t - \bar{\boldsymbol{W}}_l^0\|$ for all $1 < l \leq L$.*

Based on Lemma 4.1, we derive that feedback-weight matching reduces the network output error $e^{t+1}$ over the train step $t$ when combined with weight decay (Krogh & Hertz, 1991), as follows. The proof is detailed in Appendix A

**Proposition 4.1.** *Let $\boldsymbol{e}^t$ denote the output error of a two-layer fully connected non-linear network (i.e., L=2) at the $t$-th training step, $\eta$ is the learning rate, $\gamma \leq \lambda_{min}(\bar{\boldsymbol{G}})$ is a positive constant, where $\bar{\boldsymbol{G}} = \mathbb{E}_{w \sim \mathcal{N}(0,I_p)}\psi(w^\top x_i)\psi(w^\top x_j)$ with the number of neuron as $p$ and a non-linear function $\psi(\cdot)$, $\lambda^t$ is the weight-decay hyperparameter at the step $t$, and $\boldsymbol{y}$ is the output of the network. By applying feedback-weight matching in Equation (6) and (7), the following holds:*

$$\|\boldsymbol{e}^{t+1}\| \leq \left(1 - \frac{\eta\gamma}{4} - \eta\lambda^t\right)\|\boldsymbol{e}^t\| + \lambda^t\|\boldsymbol{y}\| - \alpha_2 r_2 \tag{13}$$

*for all $t \geq 0$ and some constants $\alpha_2$, with $r_2$ defined in (11).*

It is shown (Song et al., 2021) that the inequality in Equation (13), i.e., $\|e^{t+1}\| \leq \left(1 - \frac{\eta\gamma}{4} - \eta\lambda^t\right)\|e^t\| + \lambda^t\|y\|$, holds for a two-layer fully connected non-linear network when applying FA (Feedback Alignment) (Lillicrap et al., 2016) with weight decay (Krogh & Hertz, 1991). Specifically, the right-hand side of the inequality, i.e., $\left(1 - \frac{\eta\gamma}{4} - \eta\lambda^t\right)\|e^t\| + \lambda^t\|y\|$, consists of the following term as a linear component in fine-tuning:

$$\|W_2^t - W_2^0\| \text{ s.t. } W_2^0 \propto F_1^\top \tag{14}$$

where $W_2^0$ is the pre-trained weights. By assuming that $W_2^0$ is replaced with the re-initialized weights, $\hat{W}_2^0$ in Equation (7), $\|e^{t+1}\|$ in Equation (13) is decreased by $\alpha_2 r_2$ since $\|W_2^t - W_2^0\| \geq \|W_2^t - \hat{W}_2^0\|$, as in Lemma 4.1.

**Conjecture 4.2.** *Given an $L$-layer fully connected non-linear network, suppose that the right-hand side of the inequality in Equation (13), i.e., $\left(1 - \frac{\eta\gamma}{4} - \eta\lambda^t\right)\|e^t\| + \lambda^t\|y\|$, contains $\|W_l^t - W_l^0\|$ as linear components for some $1 < l \leq L$. Then, based on Proposition 4.1 and Lemma 4.1, it is conjectured that Equation (13) can be generalized into:*

$$\|e^{t+1}\| \leq \left(1 - \frac{\eta\gamma}{4} - \eta\lambda^t\right)\|e^t\| + \lambda^t\|y\| - \sum_{l=2}^{L} \alpha_l r_l \tag{15}$$

*with constants $\alpha_l$, and $r_l$ defined in (11) and (12) for some $1 < l \leq L$ and all $t \geq 0$.*

From Equation (13), and subsequently Equation (15), it can be seen that feedback-weight matching preserves the weight decay effect by decreasing the network error $\|e^{t+1}\|$ by the quantity $\eta\lambda^t\|e^t\| - \lambda^t\|y\|$. It is achieved by $\sum_{l=2}^{L} \alpha_l r_l$, which effectively counteracts the adverse impact of weight decay, namely, the increase in error $\|e^{t+1}\|$ when $\eta\|e^t\| \leq \|y\|$, if $\sum_{l=2}^{L} \alpha_l r_l \geq \lambda^t\|y\| - \eta\lambda^t\|e^t\|$.

## 5 EXPERIMENT

We evaluate the proposed feedback-weight matching on two types of fine-tuning tasks. First, it is applied to image classification tasks using two fully connected networks with four and six hidden layers, respectively. These networks are pre-trained with CIFAR-100 (Krizhevsky et al., 2009) and TinyImageNet (Le & Yang, 2015) using back-propagation (Rumelhart et al., 1986), and then fine-tuned on CIFAR-10 (Krizhevsky et al., 2009), SVHN (Netzer et al., 2011), and STL-10 (Coates et al., 2011) through DFA applying feedback-weight matching. Next, we apply it to natural language processing (NLP) tasks using Transformer models, i.e., BERT-Tiny and Small (Kenton & Toutanova, 2019; Turc et al., 2019), pre-trained on BookCorpus (Zhu et al., 2015) & Wikipedia, and then fine-tuned with a set of GLUE tasks (Wang, 2018). For fine-tuning of BERT, feedback-weight matching is applied to the attention, intermediate, and block outputs of the encoder layers in a similar way to previous works (Launay et al., 2020) that attempt to apply DFA to Transformer's attention architectures (Vaswani, 2017). It is important to highlight that even standard DFA has rarely been applied to Transformer models for from-scratch training due to its inherent challenges and difficulties (Launay et al., 2020). Our experiment is the first attempt to apply DFA fine-tuning to Transformer models (i.e., BERT), which is considered more challenging than from-scratch DFA training. The detailed experimental setups are provided in Appendix E.

### 5.1 FINE-TUNING PERFORMANCE

Table 1 summarizes the fine-tuning performance on image classification tasks (i.e., test classification accuracy) of the proposed feedback-weight matching compared against 1) back-propagation-based fine-tuning and 2) standard DFA fine-tuning that does not apply feedback-weight matching. As shown in the table, the proposed feedback-weight matching enables reliable fine-tuning for various network architectures and tasks, which consistently outperforms standard DFA with an average of 2.16% accuracy gap, while underperforming when compared to back-propagation with an average of 2.32%. For instance, feedback-weight matching achieves 82.67% accuracy when fine-tuning the 6-layer network from CIFAR-100 to SVHN, which is 7.97% higher than standard DFA that achieves 74.70%, but 1.67% lower than back-propagation. It also indicates that the proposed feedback-weight matching maintains more robust performance over network depths, whereas the performance of standard DFA deteriorates with deeper networks. For instance, in the case of fine-tuning from CIFAR-100 to SVHN, the accuracy drop between 4-layer and 6-layer networks is only 0.20% with feedback-weight matching, which is 24x smaller than the case not applying it (4.85%).

**Table 1: Image Classification Tasks.** The fine-tuning performance of feedback-weight matching (DFA$_{ours}$) on the 4 and 6-layer fully connected networks, compared with standard DFA fine-tuning (DFA$_{fine}$) and back-propagation fine-tuning (BP$_{fine}$). The pre-trained weights are obtained through back-propagation (BP). For reference, we also present the from-scratch-training results of back-propagation (BP$_{scratch}$) and DFA (DFA$_{scratch}$). The bold indicates better performance in DFA fine-tuning.

| Model | Target Data | Source Data | | | | | | | |
| | | Scratch | | CIFAR-100 | | | TinyImageNet | | |
| | | BP$_{scratch}$ | DFA$_{scratch}$ | BP$_{fine}$ | DFA$_{fine}$ | DFA$_{ours}$ | BP$_{fine}$ | DFA$_{fine}$ | DFA$_{ours}$ |
|---|---|---|---|---|---|---|---|---|---|
| 4 layers | CIFAR-10 | 55.48 | 52.78 | 57.16 | 53.79 | **55.38** | 57.66 | **56.75** | 55.51 |
| | SVHN | 85.10 | 82.93 | 84.32 | 79.55 | **82.87** | 84.69 | 80.31 | **83.16** |
| | STL-10 | 43.15 | 42.20 | 47.73 | 44.83 | **45.30** | 50.29 | **50.62** | 45.61 |
| 6 layers | CIFAR-10 | 54.93 | 51.94 | 58.85 | 53.04 | **55.39** | 55.97 | 51.08 | **55.54** |
| | SVHN | 85.10 | 81.89 | 84.34 | 74.70 | **82.67** | 84.72 | 76.03 | **81.39** |
| | STL-10 | 43.10 | 40.48 | 47.78 | 43.42 | **45.28** | 47.63 | 43.33 | **45.21** |

Table 2 presents the evaluation results of feedback-weight matching applied to BERT-Tiny and BERT-Small, fine-tuned for NLP tasks, using the same experimental setup in image classification tasks (Table 1). Similar to image classification tasks, feedback-weight matching enables DFA to fine-tune BERT for various tasks of the GLUE dataset in a more robust manner compared to standard DFA. For example, on CoLA, feedback-weight matching achieves a Matthews correlation of 0.53 in BERT-Small, compared to 0.06 with standard DFA. Similarly, on STSB, BERT-Small achieves a Pearson correlation of 0.76 with feedback-weight matching, while standard DFA yields only 0.10, demonstrating a significant gap in both learning performance and reliability. In the worst case, standard DFA fails to learn from the fine-tuning data entirely, achieving 0.00 Matthews correlation for CoLA with BERT-Tiny, whereas feedback-weight matching achieves 0.29.

**Table 2: NLP Tasks.** The fine-tuning performance of feedback-weight matching (DFA$_{ours}$) on Transformer architectures (i.e., BERT-Tiny and BERT-Small), compared with standard DFA fine-tuning (DFA$_{fine}$) and back-propagation-based fine-tuning (BP$_{fine}$). The pre-trained weights are obtained via back-propagation (BP). For reference, we also present the from-scratch-training results of back-propagation (BP$_{scratch}$) and DFA (DFA$_{scratch}$). The bold indicates better performance in DFA fine-tuning.

| Model | Training | CoLA (mat-cor) | SST-2 (acc) | MRPC (acc) | QQP (acc) | MNLI (acc) | QNLI (acc) | STSB (pearson) | RTE (acc) | WNLI (acc) |
|---|---|---|---|---|---|---|---|---|---|---|
| BERT-Tiny | BP$_{scratch}$ | 0.07 | 96.3 | 67.4 | 82.8 | 63.4 | 89.2 | -0.19 | 64.1 | 50.0 |
| | BP$_{fine}$ | 0.00 | 93.5 | 70.7 | 86.9 | 73.8 | 88.2 | -0.25 | 60.3 | 52.6 |
| | DFA$_{scratch}$ | 0.00 | 95.2 | 67.4 | 81.2 | 59.2 | 84.2 | -0.11 | 50.2 | 50.0 |
| | DFA$_{fine}$ | 0.00 | 92.4 | 67.4 | 80.6 | 60.0 | 80.2 | -0.17 | 51.2 | 51.0 |
| | DFA$_{ours}$ | **0.29** | **95.9** | **69.7** | **82.3** | **60.2** | **84.3** | **0.36** | **55.5** | **52.6** |
| BERT-Small | BP$_{scratch}$ | 0.55 | 96.3 | 95.4 | 91.3 | 75.3 | 93.4 | 0.67 | 89.8 | 51.9 |
| | BP$_{fine}$ | 0.87 | 98.9 | 96.7 | 98.0 | 93.0 | 99.1 | 0.90 | 94.0 | 53.3 |
| | DFA$_{scratch}$ | 0.19 | 96.5 | 75.2 | 86.7 | 67.4 | 80.9 | 0.05 | 60.0 | 50.3 |
| | DFA$_{fine}$ | 0.06 | 95.6 | 70.9 | 86.0 | **67.0** | 85.3 | 0.10 | 59.0 | 49.3 |
| | DFA$_{ours}$ | **0.53** | **97.3** | **92.5** | **86.9** | 65.8 | **87.2** | **0.76** | **59.0** | **51.0** |

## 5.2 Weight Alignment (WA) and Gradient Alignment (GA)

Figure 1a and 1b plot the weight alignment (WA) and the gradient alignment (GA), along with the train and test accuracy, for some fine-tuning setups. As shown in the figures, the proposed feedback-weight matching (green) induces strong weight alignment (WA) from the outset, subsequently strong gradient alignment (GA) as analyzed in Section 3.2 and 3.3, leading to both enhanced train and test accuracy across all experiments with faster and stable convergence. In contrast, standard DFA (yellow), not applying feedback-weight matching, achieves significantly lower WA and GA. While they gradually increase over fine-tuning epochs in some cases, the initially low WA and GA impede effective fine-tuning. As a result, the train and test accuracy of standard DFA do not improve substantially from the pre-trained weight parameters, especially for BERT-Small. This suggests that standard DFA struggles to adapt to the target dataset during fine-tuning, likely due to the mismatch between its random feedback matrices and the pre-trained weights. In other words, it overly relies on pre-trained weights in the hope that they will fit and perform well on new target fine-tuning data.

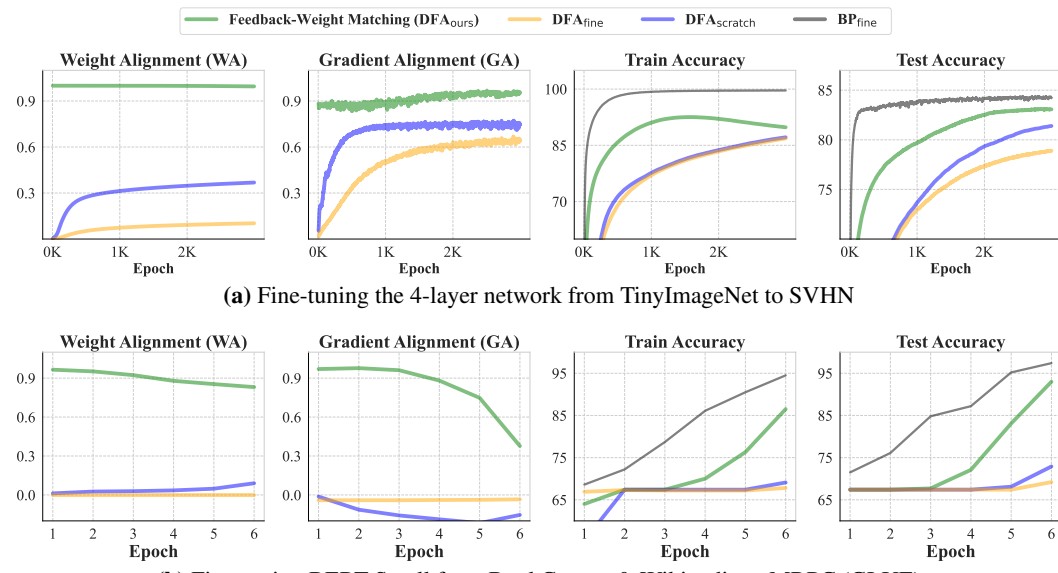

**(a)** Fine-tuning the 4-layer network from TinyImageNet to SVHN

**(b)** Fine-tuning BERT-Small from BookCorpus & Wikipedia to MRPC (GLUE)

**Figure 1: WA, GA, train accuracy, and test accuracy.** The green graph indicates DFA fine-tuning with feedback-weight matching (ours), the yellow indicates DFA fine-tuning without feedback-weight matching, the blue indicates DFA trained from scratch, and the gray indicates fine-tuning with back-propagation.

**Table 3: Ablation experiment.** The fine-tuning performance when removing weight matching ($DFA_{weight*}$), feedback matching ($DFA_{feed*}$), and weight decay ($DFA_{decay*}$). '$DFA_{ours}$' denotes applying all of them.

| Model | Target Data | Source Data | | | | | | | |
|---|---|---|---|---|---|---|---|---|---|
| | | CIFAR-100 | | | | TinyImageNet | | | |
| | | $DFA_{weight*}$ | $DFA_{feed*}$ | $DFA_{decay*}$ | $DFA_{ours}$ | $DFA_{weight*}$ | $DFA_{feed*}$ | $DFA_{decay*}$ | $DFA_{ours}$ |
| 4 layers | CIFAR-10 | 53.92 | 55.23 | 48.82 | **55.38** | 53.73 | 55.05 | 48.66 | **55.51** |
| | SVHN | 80.65 | 81.34 | 77.99 | **82.87** | 79.77 | 83.13 | 77.63 | **83.16** |
| | STL-10 | 44.25 | 45.20 | 40.00 | **45.30** | 44.05 | 45.42 | 40.47 | **45.61** |
| 6 layers | CIFAR-10 | 53.47 | 55.03 | 46.21 | **55.39** | 53.50 | 55.03 | 45.77 | **55.54** |
| | SVHN | 79.70 | 82.76 | 76.71 | **82.67** | 79.77 | 82.76 | 76.76 | **82.72** |
| | STL-10 | 43.86 | **45.42** | 39.17 | 45.28 | 43.78 | **45.43** | 40.23 | 45.21 |

## 5.3 ABLATION STUDY: FEEDBACK MATCHING, WEIGHT MATCHING, AND WEIGHT DECAY

Table 3 presents the impact of feedback matching, weight matching, and weight decay on fine-tuning with DFA. To assess their effectiveness, we remove each of them in isolation. Removing feedback matching results in a marginal performance decline, such as a reduction from 55.54% to 55.03% when the 6-layer network is fine-tuned from TinyImageNet to CIFAR-10. This marginal drop occurs because bypassing feedback matching applies random feedback matrices to the re-initialized weights that are amenable to arbitrary random feedback matrices, resulting in a reasonable level of WA and GA. In contrast, omitting weight matching leads to a relatively bigger performance drop, e.g., classification accuracy decreases from 83.16% to 79.77% when fine-tuning the 4-layer network from TinyImageNet to SVHN. Similarly, the correlation score drops from 0.76 to -0.06 when fine-tuning BERT-Small to STSB as shown in Table 5 (Appendix D). It is presumed that excluding weight matching causes the pre-trained weights obtained by back-propagation, not by DFA, to be fine-tuned with mismatched feedback matrices, thereby resulting in weak WA and GA.

When weight decay is not applied, the fine-tuning of feedback-weight matching performance also exhibits some declines, e.g., classification accuracy decreases from 55.38% to 48.82% when fine-tuning the 4-layer network from CIFAR-100 to CIFAR-10. It should be noted that weight decay appears to have minimal impact on fine-tuning of standard DFA when feedback-weight matching is not applied; in our experiment, the classification accuracy even increases, such as from 54.38% to 56.75% when fine-tuning the 4-layer network from TinyImageNet to CIFAR-10. This demonstrates the synergistic effect of feedback-weight matching and weight decay, i.e., reducing network output error as shown in Section 4.

## 5.4 FEEDBACK-WEIGHT MATCHING AND WEIGHT DECAY

To evaluate the impact of feedback-weight matching on weight decay, we measure the fine-tuning performance with weight decay, with and without applying feedback-weight matching, which is shown in Table 4. The results indicate that weight decay enhances fine-tuning accuracy (reducing network output error) when used in conjunction with feedback-weight matching, with an average improvement of 8.35%. This demonstrates that feedback-weight matching facilitates weight decay in reducing network output error, thereby improving fine-tuning accuracy, as provided in Equation (15). In contrast, weight decay is less likely to improve fine-tuning performance without feedback-weight matching. In fact, when applied to the standard DFA (not applying feedback-weight matching), weight decay results in fine-tuning accuracy with minimal variation (providing similar accuracy).

**Table 4: Feedback-weight matching and weight decay .** 'DFA$_{fine}$' applies weight decay without feedback-weight matching, compared with 'DFA$_{ours}$' applying both weight decay and feedback-weight matching.

**(a)** Fine-tuning image classification tasks (fully connected networks)

| Target Data | 4 layers | | | | 6 layers | | | |
|---|---|---|---|---|---|---|---|---|
| | CIFAR-100 | | TinyImageNet | | CIFAR-100 | | TinyImageNet | |
| | DFA$_{fine}$ | DFA$_{ours}$ | DFA$_{fine}$ | DFA$_{ours}$ | DFA$_{fine}$ | DFA$_{ours}$ | DFA$_{fine}$ | DFA$_{ours}$ |
| CIFAR-10 | 54.39 | **55.38** | 54.38 | **55.51** | 54.08 | **55.39** | 53.50 | **55.54** |
| SVHN | 80.77 | **82.87** | 80.74 | **83.16** | 78.73 | **82.67** | 79.57 | **82.72** |
| STL-10 | 45.00 | **45.30** | **50.40** | 45.61 | 43.56 | **45.28** | **45.28** | 45.21 |

**(b)** Fine-tuning NLP tasks (BERT)

| Model | Training | CoLA (mat-cor) | SST-2 (acc) | MRPC (acc) | QQP (acc) | MNLI (acc) | QNLI (acc) | STSB (pearson) | RTE (acc) | WNLI (acc) |
|---|---|---|---|---|---|---|---|---|---|---|
| BERT-Tiny | DFA$_{fine}$ | 0.00 | 92.4 | 67.4 | 80.6 | 60.0 | 80.2 | -0.17 | 51.2 | 51.0 |
| | DFA$_{ours}$ | **0.29** | **95.9** | **69.7** | **82.3** | **60.2** | **84.3** | **0.36** | **55.5** | **52.6** |
| BERT-Small | DFA$_{fine}$ | 0.06 | 95.6 | 70.9 | 86.0 | **67.0** | 85.3 | 0.10 | 59.0 | 49.3 |
| | DFA$_{ours}$ | **0.53** | **97.3** | **92.5** | **86.9** | 65.8 | **87.2** | **0.76** | **59.0** | **51.0** |

Figure 2 plots the weight alignment (WA), gradient alignment (GA), training accuracy, and test accuracy across varying strengths of weight decay during the fine-tuning of 4-layer network from CIFAR-100 to CIFAR-10. The proposed feedback-weight matching ensures strong WA and GA as discussed in Section 3.2 and 3.3 from the beginning, which helps mitigate alignment degradation (Song et al., 2021), while exhibiting varying behaviors depending on different levels of weight decay. In the absence of weight decay (black curve), GA declines and exhibits significant oscillations, ultimately causing a decrease in test accuracy. Conversely, when a strong weight decay is applied (blue curve), both WA and GA decrease sharply, followed by substantial reductions in both training and test accuracy. These observations suggest that an appropriate weight decay strength is crucial for effective fine-tuning (green curve) when applied with feedback-weight matching.

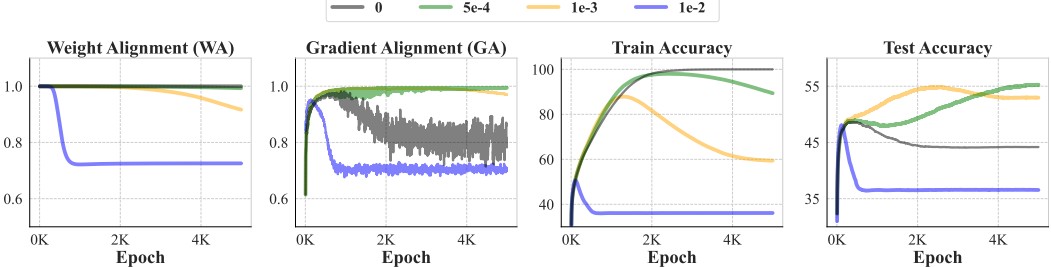

**Figure 2: WA, GA, train accuracy, and test accuracy** over different weight decays (0, 5e-4, 1e-3, and 1e-2). A 4-layer fully connected network is fine-tuned from CIFAR-100 to CIFAR-10 by feedback-weight matching.

## 6 LIMITATIONS AND FUTURE WORKS

We discuss the limitations and future works of this paper in Appendix C.

## 7 CONCLUSION

We propose feedback-weight matching, a method that enhances the fine-tuning capability and stability of Direct Feedback Alignment (DFA) for pre-trained networks. While standard DFA struggles in fine-tuning networks trained via back-propagation, the proposed feedback-weight matching improves weight and gradient alignment, boosting stability and performance of DFA fine-tuning. Combined with weight decay, it also reduces over-fitting and network errors. Our experiments show significant improvements in both image classification and NLP tasks compared to standard DFA.

## 8 REPRODUCIBILITY STATEMENT

For reproduction of the experimental results presented in this paper, we provide access to an anonymous GitHub repository[1] containing the code implementation and reproduction instructions. The detailed experimental setups are provided in Appendix E.

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

# A  PROOF

## A.1  PROOF OF PROPOSITION 3.3

*Proof.* We prove Proposition 3.3 for $W_{1<l<L}^t$ in Equation (16), and the same reasoning applies to $W_L^t$ in (17). Since $A_{l\geq 2}^t$ in Equation (3) becomes such that $A_{l\geq 2}^t \propto I$ as the training proceeds (Refinetti et al., 2021), the weight newly updated with DFA, which is denoted as $\bar{W}_{1<l<L}^t$, comes to satisfy Equation (4), i.e., $\bar{W}_{1<l<L}^t = c_l^t F_l F_{l-1}^\top$ with some constant $c_l^t$. Given that we take the pre-trained weight $W_{1<l<L}^0$ as the initial point in our fine-tuning, the overall weight $W_{1<l<L}^t$ obtained by DFA is expressed as the sum of $W_{1<l<L}^0$ and $\bar{W}_{1<l<L}^t$, which is given by:

$$W_{1<l<L}^t = W_{1<l<L}^0 + \bar{W}_{1<l<L}^t = W_{1<l<L}^0 + c_l^t F_l F_{l-1}^\top \not\propto F_l F_{l-1}^\top \qquad (16)$$

$$W_L^t = W_L^0 + \bar{W}_L^t = W_L^0 + c_L^t F_{L-1}^\top \not\propto F_{L-1}^\top \qquad (17)$$

where $c_{1<l\leq L}^t$ is a constant. In Equation (16), since $W_{1<l<L}^0$ is unlikely to be proportional to $F_l F_{l-1}^\top$, i.e., $W_{1<l<L}^0 \not\propto F_l F_{l-1}^\top$, the overall weight $W_{1<l<L}^t$, which includes $W_{1<l<L}^0$, is also unlikely to be proportional to $F_l F_{l-1}^\top$, i.e., $W_{1<l<L}^t \not\propto F_l F_{l-1}^\top$, though $\bar{W}_{1<l<L}^t = c_l^t F_l F_{l-1}^\top \propto F_l F_{l-1}^\top$. Hence, Equation (16) can hardly induce strong WA in Equation (4). $\square$

## A.2  PROOF OF PROPOSITION 3.6

*Proof.* Similar to (16) and (17), the overall weight $W_l^t$ obtained by DFA is the sum of $W_l^0$ and $\bar{W}_l^t$. Specifically, now that $\bar{W}_{1<l<L}^0 = \bar{F}_l \bar{F}_{l-1}^\top$ and $\bar{W}_L^0 = \bar{F}_{L-1}^\top$, these become proportional to $\bar{F}_l \bar{F}_{l-1}^\top$ and $\bar{F}_{L-1}$, respectively, as follows:

$$W_{1<l<L}^t = \bar{W}_{1<l<L}^0 + \bar{W}_{1<l<L}^t = \bar{F}_l \bar{F}_{l-1}^\top + c_l^t \bar{F}_l \bar{F}_{l-1}^\top = (1+c_l^t)\bar{F}_l \bar{F}_{l-1}^\top \propto \bar{F}_l \bar{F}_{l-1}^\top \qquad (18)$$

$$W_L^t = \bar{W}_L^0 + \bar{W}_L^t = \bar{F}_{L-1}^\top + c_L^t \bar{F}_{L-1}^\top = (1+c_L^t)\bar{F}_{L-1}^\top \propto \bar{F}_{L-1}^\top \qquad (19)$$

with constants $c_{1<l\leq L}^t$, which aligns with the strong WA condition in Equation (4). $\square$

## A.3  PROOF OF PROPOSITION 3.8

*Proof.* The weight at the second layer of the network, $W_2^t$, can be expressed with the pre-trained weight, $W_2^0$, with the learning rate $\eta$, the number of neurons as $p$, $F_1 \in \mathbb{R}^p$, and $W_2^t \in \mathbb{R}^p$ as follows (Song et al., 2021).

$$W_2^t = W_2^{t-1} - \eta \frac{1}{\sqrt{p}} W_1^{t-1} X^\top e^{t-1} = W_2^0 - \frac{\eta}{\sqrt{p}} \sum_{t=0}^{t'-1} W_1^t X^\top e^t \qquad (20)$$

For the standard DFA that does not apply feedback-weight matching in Equation (6) and (7), we have $G_{DFA} = F_1$ and $G_{BP} = W_2^t$. By using Equation (20), the gradient alignment (GA) defined in Equation (9) between them, which is denoted as $\cos_{DFA} \angle(F_1, W_2^t)$, is at least as follows.

$$\cos_{DFA} \angle(F_1, W_2^t) = \frac{F_1^\top W_2^t}{\|F_1\| \|W_2^t\|} = \frac{\frac{F_1^\top}{\|F_1\|} W_2^t}{\|W_2^t\|} = \frac{\frac{F_1^\top}{\|F_1\|}(W_2^0 - \frac{\eta}{\sqrt{p}} \sum_{t=0}^{t'-1} W_1^t X^\top e^t)}{\|W_2^0 - \frac{\eta}{\sqrt{p}} \sum_{t=0}^{t'-1} W_1^t X^\top e^t\|}$$

$$\geq \frac{\frac{F_1^\top}{\|F_1\|}(W_2^0 - \frac{\eta}{\sqrt{p}} \sum_{t=0}^{t'-1} W_1^t X^\top e^t)}{\|W_2^0\| + \|\frac{\eta}{\sqrt{p}} \sum_{t=0}^{t'-1} W_1^t X^\top e^t\|} \qquad (21)$$

Conversely, when applying feedback-weight matching in Equation (6) and (7), we have $F_1 = W_2^0$ for $L=2$. Using Equation (20) again, GA between them, $\cos_{FWM} \angle(F_1, W_2^t)$, is at least as follows.

$$\cos_{FWM} \angle(F_1, W_2^t) = \frac{\frac{F_1^\top}{\|F_1\|}(W_2^0 - \frac{\eta}{\sqrt{p}} \sum_{t=0}^{t'-1} W_1^t X^\top e^t)}{\|W_2^0 - \frac{\eta}{\sqrt{p}} \sum_{t=0}^{t'-1} W_1^t X^\top e^t\|} \geq \frac{\frac{F_1^\top}{\|F_1\|}(F_1 - \frac{\eta}{\sqrt{p}} \sum_{t=0}^{t'-1} W_1^t X^\top e^t)}{\|F_1\| + \|\frac{\eta}{\sqrt{p}} \sum_{t=0}^{t'-1} W_1^t X^\top e^t\|}.$$

$$(22)$$

If we assume that both $F_1$ and $W_2^0$ follow the standard Gaussian distribution, we have $\|F_1^\top W_2^0\| \leq \|F_1\|^2$ (Song et al., 2021). Thus, $\cos_{FWM} \angle(F_1, W_2^t)$ exhibits a higher lower bound compared to $\cos_{DFA} \angle(F_1, W_2^t)$, i.e., $\cos_{FWM} \angle(F_1, W_2^t) \geq \cos_{DFA} \angle(F_1, W_2^t)$, implying a higher GA. $\square$

### A.4 PROOF OF LEMMA 4.1

*Proof.* We show that $r_{1 \leq l < L} \geq 0$ in Equation (11), and the same reasoning extends to $r_L$ in (12). Given that $\bar{W}_l^0 = \bar{F}_l \bar{F}_{l-1}^\top \propto W_l^t = c_l^t \bar{F}_l \bar{F}_{l-1}^\top$, we can interpret $W_l^t$ as a scaled version of $\bar{W}_l^0$, which implies that $\|W_l^t - \bar{W}_l^0\|$ is small. Conversely, since $W_l^0$ is not proportional to $W_l^t$, i.e., $W_l^0 \not\propto W_l^t = c_l^t \bar{F}_l \bar{F}_{l-1}^\top$, it follows that $\|W_l^t - W_l^0\|$ is generally larger than $\|W_l^t - \bar{W}_l^0\|$. Therefore, $\|W_l^t - \bar{W}_l^0\|$ is likely smaller than $\|W_l^t - W_l^0\|$. $\qquad \square$

## B DECOMPOSITION OF WEIGHT INTO FEEDBACK MATRICES

One way of finding feedback matrices $\bar{F}_l$ and $\bar{F}_{l-1}^\top$ in Equation (6) from $W_{1 < l < L}^0$, other than SVD (Singular Value Decomposition) (Klema & Laub, 1980), is to optimize the following objective $\mathcal{L}_{FM}$.

$$\mathcal{L}_{FM} = \frac{1}{2} \sum_{l=2}^{L-1} (W_l^0 h_{l-1} - \bar{F}_l \bar{F}_{l-1}^\top h_{l-1})^2 + \frac{1}{2} (W_L^0 h_{L-1} - \bar{F}_{L-1} h_{L-1})^2 + \frac{1}{2} \sum_{l=1}^{L-1} (I - \bar{F}_l^\top \bar{F}_l)^2 \tag{23}$$

Here, $\mathcal{L}_{FM}$ is minimized to ensure that the layer output, when replaced by the feedback matrix $\bar{F}_l \bar{F}_{l-1}^\top h_{l-1}$, matches the output obtained using the pre-trained weight $W_l^0 h_{l-1}$, while $\bar{F}_l$ is to be orthogonal to itself in accordance with the regular DFA condition (Lillicrap et al., 2016).

## C LIMITATIONS AND FUTURE WORKS

**Extending to Different Architectures**. Although this study presents the significant potential of fine-tuning with DFA, its current application is restricted to fully connected networks. This limitation arises because, at present, DFA is predominantly effective for fully connected architectures, and further research is needed to extend its applicability to other network types. In our future work, we plan to explore the application of DFA fine-tuning to various network architectures, such as CNNs. Meanwhile, we anticipate the development of more generalized methods that will enable DFA to be applied across a broader range of network types, thereby enhancing the applicability of our work.

**Improving Learning Performance**. The learning performance of the proposed feedback-weight matching is shown to surpass both 1) training networks with DFA from scratch and 2) fine-tuning networks with DFA using random feedback matrices. While fine-tuning with DFA applying the proposed method achieves superior and more stable performance compared to them, it still falls short of the performance achieved with fine-tuning using back-propagation (Rumelhart et al., 1986). We plan to explore how to achieve fine-tuning performance comparable to that of back-propagation by investigating DFA from its fundamental mechanism, along with the proposed method.

**Proving Hypotheses**. This work provides some hypotheses regarding fine-tuning and weight decay in the context of DFA. For example, Conjecture 3.9 suggests that applying the proposed feedback-weight matching can achieve strong weight alignment (WA) for fully connected networks of arbitrary depth. Additionally, Conjecture 4.2 posits that applying the proposed method to weight decay enhances fine-tuning performance of DFA for fully connected networks of arbitrary layers. However, formal proofs are necessary to substantiate these hypotheses and validate the efficacy of the proposed approach. In future research, we intend to generalize the propositions presented in this study to encompass various types of fully connected network architectures.

## D ABLATION EXPERIMENT ON BERT

Table 5 presents the fine-tuning performance of BERT models when weight matching, feedback matching, and weight decay are individually removed. It is important to note that DFA is not applied to all fully connected layers in BERT, which limits the ability to properly assess the effectiveness of feedback-weight matching. Thus, this experimental setup may not provide an accurate evaluation.

## E EXPERIMENTAL SETUPS

In this section, we offer an explanation of the experimental setup utilized throughout our research. Appendix E.1 outlines the training details of the feedback matrix used for feedback matching in all

**Table 5: Ablation experiment.** The fine-tuning performance when removing weight matching (DFA$_{weight*}$), feedback matching (DFA$_{feed*}$), and weight decay (DFA$_{decay*}$). 'DFA$_{ours}$' denotes applying all of them.

| Model | Training | CoLA (mat-cor) | SST-2 (acc) | MRPC (acc) | QQP (acc) | MNLI (acc) | QNLI (acc) | STSB (pearson) | RTE (acc) | WNLI (acc) |
|---|---|---|---|---|---|---|---|---|---|---|
| BERT-Tiny | DFA$_{weight*}$ | 0.00 | 94.7 | 67.4 | 81.4 | 59.2 | 88.4 | -0.15 | 50.3 | 50.9 |
| | DFA$_{feed*}$ | 0.00 | 95.8 | 68.9 | 82.4 | 60.8 | 86.9 | 0.35 | 55.5 | 50.0 |
| | DFA$_{decay*}$ | 0.31 | 95.9 | 71.4 | 81.9 | 61.0 | 83.3 | 0.36 | 53.3 | 51.9 |
| | DFA$_{ours}$ | 0.29 | 95.9 | 69.7 | 82.3 | 60.2 | 84.3 | 0.36 | 50.8 | 52.6 |
| BERT-Small | DFA$_{weight*}$ | 0.08 | 96.0 | 75.1 | 85.0 | 66,7 | 79.7 | -0.06 | 61.8 | 50.1 |
| | DFA$_{feed*}$ | 0.54 | 97.0 | 91.5 | 87.4 | 65.2 | 85.3 | 0.75 | 62.0 | 50.2 |
| | DFA$_{decay*}$ | 0.53 | 97.2 | 91.2 | 87.1 | 64.7 | 85.4 | 0.78 | 68.7 | 50.9 |
| | DFA$_{ours}$ | 0.53 | 97.3 | 92.5 | 86.9 | 65.8 | 87.2 | 0.76 | 59.0 | 51.0 |

models. Appendix E.2 covers the configuration settings required for the fully connected network experiments. Appendix E.3 describes the setup necessary for experiments involving BERT, which employs a transformer architecture. To ensure the robustness of our findings, we report the average results over three different random seeds.

### E.1 FEEDBACK MATRIX

We train feedback matrices to reconstruct pre-trained weights that were trained using back-propagation (Rumelhart et al., 1986). The loss function, in Equation (23), is used to guide the feedback matching process. The two learned feedbacks are then combined and re-initialized into a single weight matrix for each layer. We use the Adam optimizer (Kingma, 2014) without weight decay or any scheduler. In fully connected networks, a learning rate of 1e-5 is applied, while in transformers (BERT) (Kenton & Toutanova, 2019; Turc et al., 2019), a learning rate of 1e-3 is used. For all experiments on the model and dataset, training is conducted for 3 epochs with a batch size of 64.

### E.2 FULLY CONNECTED NETWORKS

We pre-train two fully connected networks with four and six layers on the CIFAR-100 (Krizhevsky et al., 2009) and TinyImageNet (Le & Yang, 2015) datasets utilizing weights obtained through back-propagation (BP). These pre-trained weights are subsequently fine-tuned on the CIFAR-10 (Krizhevsky et al., 2009), SVHN (Netzer et al., 2011), and STL-10 (Coates et al., 2011) datasets. During the pre-processing phase, we apply image resizing and normalization, without any augmentations. For Dynamic Feedback Alignment (DFA) (Nøkland, 2016), the weights are initialized with a uniform distribution within the range of (-0.01, 0.01). Conversely, for back-propagation (Rumelhart et al., 1986), we employ the He initialization (He et al., 2015). The optimization process is carried out using Stochastic Gradient Descent, and ReLU (Agarap, 2018) is employed as the activation function. The hyperparameters for both the 4-layer and 6-layer architectures remain consistent. A comprehensive description of each hyperparameter under various training conditions is presented in Table 6.

**Table 6: Hyperparameters for fully connected networks training.**

| Target Data | Hyperparmeters | BP$_{scratch}$ | BP$_{fine}$ | DFA$_{scratch}$ | DFA$_{fine}$ | DFA$_{feed}$ | DFA$_{weight}$ | DFA$_{ours}$ |
|---|---|---|---|---|---|---|---|---|
| | Learning Rate | 1e-3 | 1e-3 | 1e-3 | 1e-3 | 1e-3 | 1e-3 | 1e-3 |
| | Batch size | 64 | 64 | 64 | 64 | 64 | 64 | 64 |
| | Hidden Dim | 1000 | 1000 | 1000 | 1000 | 1000 | 1000 | 1000 |
| | Input size | 3072 | 3072 | 3072 | 3072 | 3072 | 3072 | 3072 |
| CIFAR-10 | Epochs | 5000 | 5000 | 5000 | 5000 | 5000 | 5000 | 5000 |
| | Weight Decay | 5e-4 | 5e-4 | 0 | 0 | 5e-4 | 5e-4 | 5e-4 |
| | Dropout | 0.1 | 0.1 | 0 | 0 | 0 | 0 | 0 |
| SVHN | Epochs | 5000 | 5000 | 5000 | 5000 | 5000 | 5000 | 5000 |
| | Weight Decay | 5e-4 | 5e-4 | 0 | 0 | 5e-4 | 5e-4 | 5e-4 |
| | Dropout | 0.1 | 0.1 | 0 | 0 | 0 | 0 | 0 |
| STL-10 | Epochs | 5000 | 5000 | 5000 | 5000 | 30000 | 30000 | 30000 |
| | Weight Decay | 5e-4 | 5e-4 | 0 | 0 | 1e-3 | 1e-3 | 1e-3 |
| | Dropout | 0.1 | 0.1 | 0 | 0 | 0.1 | 0.1 | 0.1 |

### E.3 BERT

We train BERT-Tiny and Small models (Kenton & Toutanova, 2019; Turc et al., 2019) on the GLUE (Wang, 2018) dataset using the AdamW (Loshchilov, 2017) optimizer with a fixed learning rate and no scheduler. We apply weight decay and dropout techniques. GeLU (Hendrycks & Gimpel, 2016) is used for the activation function, which is commonly employed in BERT. Layers such as the encoder block outputs, intermediate outputs, and attention outputs are optimized using Dynamic Feedback Alignment (DFA) (Nøkland, 2016), while the projection layers for key, query, and value are trained using back-propagation (BP) (Rumelhart et al., 1986). The weights are initialized using a uniform distribution, and the feedback matrix is specifically designed to satisfy the left orthogonality condition. A comprehensive description of the hyperparameter values is presented in Table 7.

**Table 7: Hyperparameters for BERT training.**

| Model | Hyperparmeters | Target Data | $BP_{scratch}$ | $BP_{fine}$ | $DFA_{scratch}$ | $DFA_{fine}$ | $DFA_{feed}$ | $DFA_{weight}$ | $DFA_{ours}$ |
|---|---|---|---|---|---|---|---|---|---|
| | Batch size | | 64 | 64 | 64 | 64 | 64 | 64 | 64 |
| | Dropout | | 0.1 | 0.1 | 0.1 | 0.1 | 0.1 | 0.1 | 0.1 |
| | Weight Decay | | 0.01 | 0.01 | 0.01 | 0.01 | 0.01 | 0.01 | 0.01 |
| | Epochs | | 6 | 6 | 6 | 6 | 6 | 6 | 6 |
| | Max length | | 512 | 512 | 512 | 512 | 512 | 512 | 512 |
| | Num of heads | | 2 | 2 | 2 | 2 | 2 | 2 | 2 |
| | Num of layers | | 2 | 2 | 2 | 2 | 2 | 2 | 2 |
| | Hidden dim | | 128 | 128 | 128 | 128 | 128 | 128 | 128 |
| | Intermediate dim | | 512 | 512 | 512 | 512 | 512 | 512 | 512 |
| BERT-Tiny | | CoLA | 1e-5 | 1e-5 | 1e-5 | 1e-5 | 1e-5 | 1e-5 | 1e-5 |
| | | SST-2 | 1e-5 | 1e-5 | 1e-5 | 1e-5 | 1e-5 | 1e-5 | 1e-5 |
| | | MRPC | 1e-5 | 1e-5 | 1e-5 | 1e-5 | 1e-5 | 1e-5 | 1e-5 |
| | | QQP | 1e-5 | 1e-5 | 1e-5 | 1e-5 | 1e-5 | 1e-5 | 1e-5 |
| | Learning Rate | MNLI | 1e-5 | 1e-5 | 1e-5 | 1e-5 | 1e-5 | 1e-5 | 1e-5 |
| | | QNLI | 5e-5 | 5e-5 | 5e-5 | 5e-5 | 5e-5 | 5e-5 | 5e-5 |
| | | STSB | 1e-5 | 1e-5 | 1e-5 | 1e-5 | 1e-5 | 1e-5 | 1e-5 |
| | | RTE | 1e-5 | 1e-5 | 1e-5 | 1e-5 | 1e-5 | 1e-5 | 1e-5 |
| | | WNLI | 5e-5 | 5e-5 | 5e-5 | 5e-5 | 5e-5 | 5e-5 | 5e-5 |
| | Num of heads | | 8 | 8 | 8 | 8 | 8 | 8 | 8 |
| | Num of layers | | 4 | 4 | 4 | 4 | 4 | 4 | 4 |
| | Hidden of dim | | 512 | 512 | 512 | 512 | 512 | 512 | 512 |
| | Intermediate dim | | 2048 | 2048 | 2048 | 2048 | 2048 | 2048 | 2048 |
| BERT-Small | | CoLA | 1e-5 | 1e-5 | 1e-5 | 1e-5 | 1e-5 | 1e-5 | 1e-5 |
| | | SST-2 | 1e-5 | 1e-5 | 1e-5 | 1e-5 | 1e-5 | 1e-5 | 1e-5 |
| | | MRPC | 1e-5 | 1e-5 | 1e-5 | 1e-5 | 1e-5 | 1e-5 | 1e-5 |
| | | QQP | 1e-5 | 1e-5 | 1e-5 | 1e-5 | 1e-5 | 1e-5 | 1e-5 |
| | Learning Rate | MNLI | 1e-5 | 1e-5 | 1e-5 | 1e-5 | 1e-5 | 1e-5 | 1e-5 |
| | | QNLI | 5e-5 | 5e-5 | 5e-5 | 5e-5 | 5e-5 | 5e-5 | 5e-5 |
| | | STSB | 1e-5 | 1e-5 | 1e-5 | 1e-5 | 1e-5 | 1e-5 | 1e-5 |
| | | RTE | 1e-5 | 1e-5 | 1e-5 | 1e-5 | 1e-5 | 1e-5 | 1e-5 |
| | | WNLI | 1e-5 | 1e-5 | 1e-5 | 1e-5 | 1e-5 | 1e-5 | 1e-5 |

