# OpenReview forum: "Enabling Fine-Tuning of Direct Feedback Alignment via Feedback-Weight Matching"
_ICLR.cc/2025/Conference — Submitted to ICLR 2025_

### Official Review · Reviewer_49ZY · 2024-11-03

**Soundness:** 2
**Presentation:** 2
**Contribution:** 2
**Rating:** 5
**Confidence:** 2

**Summary:**

In this paper, the author presents a feedback-weight matching strategy that amalgamates Direct Feedback Alignment (DFA) (Nøkland, 2016) and back-propagation gradients to refine the pretrained classification models. The proposed approach attains superior performance compared to DFA, yet falls short of the performance exhibited by back-propagation based techniques. The author offers an in-depth theoretical analysis and validates the proposed method using basic network architectures.

**Strengths:**

1) The author offers a comprehensive theoretical analysis.
2) The proposed method attains superior performance compared to the original Direct Feedback Alignment.

**Weaknesses:**

1) The suggested approach appears to be ineffective. It yields inferior results compared to back-propagation based methods. The suggested strategy aligns the Direct Feedback weight with the back-propagation weight, resulting in enhanced performance. In my view, the improvement in results primarily stems from the back-propagation optimization.

2) The proposed method achieves lower performance than back-propagation methods.

**Questions:**

It appears that the enhanced efficacy still stems from the back-propagation optimization. Can the author elucidate on this?

---

> ### Author Response · Authors · 2024-11-18
> **Response to Reviewer 49ZY**
>
> Thank you for your valuable comments on our work. We appreciate your detailed feedback and questions regarding the shortcomings. Below, we provide our answers to your review.
> - - -
> **Response to Q1, W1, W2:**
>
> We acknowledge this issue. Since our method relies on weights trained with back-propagation, we recognize its limitation in achieving lower performance compared to back-propagation.
>
> First, it is difficult for the DFA method to surpass the performance of back-propagation.
>
> This is a fundamental limitation of DFA, not our method. Unlike back-propagation, which tracks gradient changes layer by layer, DFA transmits a single error to all layers using random feedback matrices to approximate the gradient. This inevitably leads to performance differences. In fact, in existing papers as well, it is rare for DFA to surpass back-propagation in terms of performance.
>
> Therefore, rather than aiming to surpass back-propagation, we focused on improving the performance of DFA. Specifically, we aimed to address the issue where fine-tuning with back-propagation weights does not work well in DFA.
>
> Fine-tuning pre-trained weights with random feedback matrices often results in poor performance, as demonstrated in our experiments. In some cases, it even underperforms compared to training from random weights. In the case of training the BERT-Small model on the MRPC dataset, the accuracy is 75.2% for training from scratch, 70.9% for fine-tuning from back-propagation weights, and 92.5% for our method.
>
> This is because the random feedback matrices are entirely unrelated to the pre-trained weights. In DFA, it is the feedback matrices—not the weights—that transmits errors to each layer during the backward pass, making its role crucial.
>
> Our contribution lies in improving fine-tuning performance in DFA by considering the feedback matrix that incorporates high-performing pre-trained weights into DFA.
>
> Although not mentioned in the paper, we believe that our method would be effective in scenarios where DFA is utilized more efficiently than back-propagation, as suggested by *Launay, Julien, et al. in "Hardware beyond backpropagation: a photonic co-processor for direct feedback alignment" (arXiv preprint arXiv:2012.06373 2020)*.

---

> > ### Comment · Reviewer_49ZY · 2024-11-29
> > **Feedback for Responses**
> >
> > Thank you for resolving my doubts. I have revisited the paper and the reviews of other reviewers. I  agree with the author that it is indeed meaningful to apply DFA to fine-tuning.
> >
> > The remaining concern is with the author's statement in the related work section:
> >
> > "DFA with Weight Decay. In the study by Song et al. (2021), it is analyzed that weight decay (Krogh & Hertz, 1991) can reduce the output error in fully connected networks when used with Feedback Alignment (FA) (Lillicrap et al., 2016). Nevertheless, the analysis predominantly focuses on the training of networks from scratch using FA, rather than on the fine-tuning process with DFA.
> > "
> >
> > Apart from the difference in the initial state (randomly initialized network versus a pretrained network), what are the significant technical distinctions between the proposed DFA with Weight Decay and the method described above? This constitutes the core contribution of the proposed approach, and I suggest the author clarify it more explicitly.
> >
> > DFA, when combined with fine-tuning, achieves superior performance compared to DFA applied to models trained from scratch. The improvement may stem from two factors: 1) the proposed DFA combined with the fine-tuning strategy, and 2) the original pretrained model. This raises the initial question: "Does the enhanced efficacy derive from backpropagation optimization (i.e., pretrained models)?"
> >
> > The author should design experiments to verify which aspects contribute to the observed improvement.
> >
> > Another minor suggestion:
> > The claim for "we argue that utilizing DFA for fine-tuning would be a more effective, efficient, practical, and expedient approach" is exaggerated.
> > After all, the applying DFA to finetuning also1) consistently underperforms relative to back-propagation and 3) is limited to a narrower range of architectures.
> >
> > Based on above comments, I would keep my score. I expect the author given more detailed analysis and comparison.

---

> > > ### Author Response · Authors · 2024-11-30
> > > **Response to Reviewer 49ZY**
> > >
> > > We truly appreciate the time and effort you have dedicated, as well as the valuable constructive feedback you have shared. Your insightful comments and careful observations have been incredibly helpful in improving and refining our work. Below is the response to the feedback.
> > >
> > > ---
> > >
> > > **1. Contribution of weight decay**
> > >
> > > There are no technical differences in how weight decay is applied. While there are no technical differences in the application of weight decay, our contribution lies in proving its effectiveness within the context of our method, both analytically and experimentally. Both the referenced paper and our study assert that weight decay is effective, but the proof is provided in different situations. The referenced paper analytically demonstrates that weight decay improves performance when training from scratch with Feedback Alignment (FA).
> > >
> > > In contrast, our method neither trains from scratch nor employs FA. Therefore, the analysis of the referenced paper cannot confirm whether weight decay would be effective in our approach. To address this gap, we provided both theoretical and empirical evidence to demonstrate the effectiveness of weight decay in our methodology.
> > > Moreover, in DFA fine-tuning method that directly reuses back-propagation, weight decay does not lead significant performance improvements. In contrast, our method, Feedback-weight Matching, demonstrates that weight decay leads to significant performance improvements, as shown in Table 3 and 4. This suggests that, in our method, weight decay not only helps address general overfitting but also contributes significantly to fine-tuning performance.
> > >
> > > In conclusion, our contribution is twofold: 1) we apply weight decay to our method, Feedback-weight Matching, and 2) we provide both experimental and theoretical evidence that weight decay not only prevents general overfitting but also improves performance in fine-tuning.
> > >
> > > **2. Impacted by back-propagation**
> > >
> > > We believe the experimental results demonstrate the distinction between $DFA_{fine}$ and $DFA_{ours}$ in Table 1 and 2. In the case of $DFA_{fine}$, the weights from back-propagation are directly reused, whereas $DFA_{ours}$ utilized feedback matrices derived from back-propagation to initialize the weights. If the performance improvement is primarily driven by back-propagation weights, $DFA_{fine}$ would show consistently performance improvements. However, its performance is lower in some cases.
> > >
> > > Our method is inspired by back-propagation but focuses on leveraging the feedback matrix to make it applicable within DFA. Therefore, it cannot be concluded that our approach relies entirely on the back-propagation weights (pre-trained weights). Simply fine-tuning with pre-trained weights in DFA does not necessarily lead to performance improvements. We believe that the results obtained using our method provide evidence that enables DFA fine-tuning.
> > >
> > > **3. Suggestion for exaggerated expressions**
> > >
> > > We have revisited this statement and agree with your observation. We acknowledge that our wording may have led to some misunderstanding, and we appreciate your constructive feedback.
> > >
> > > Our intention was not to imply that DFA should replace back-propagation during fine-tuning. Rather, we aimed to argue that utilizing fine-tuning in DFA, in addition to back-propagation, would enable more efficient utilization of DFA overall. We will revise the statement as follows to better convey this intention.
> > >
> > > "We argue that utilizing fine-tuning for DFA would be a more effective, efficient, practical, and expedient approach."
> > >
> > > ---
> > >
> > > Thank you for your feedback. If there are any remaining uncertainties, we will do our best to address them within the given time.

---

> ### Comment · Area_Chair_iXBs · 2024-11-25
>
> Dear Reviewer 49ZY,
>
> Could you kindly review the rebuttal thoroughly and let us know whether the authors have adequately addressed the issues raised or if you have any further questions.
>
> Best,
>
> AC of Submission6057

---

### Official Review · Reviewer_tMBJ · 2024-11-03

**Soundness:** 3
**Presentation:** 3
**Contribution:** 3
**Rating:** 6
**Confidence:** 2

**Summary:**

This paper proposes feedback-weight matching to improve the reliability of fin-tuning for fully connected networks using direct feedback alignmeng.

**Strengths:**

+ It is quite interesting to investigate feasibility of using DFA for fine-tuning rather than training from scratch.
+ The proposed approach uses weight decay on top of feedback-weight matching to improve stability of fine-tuning, which makes sense.
+ It has been shown that the proposed approach can work on transformer models for NLP tasks, which is more challenging.

**Weaknesses:**

- It is not clear why the proposed approach hasn't been tested for other transformer models like ViT?
- The proposed approach has been evaluated on a limited number of datasets. Can it work on models trained, or do fine-tuning on imagenet?
- Some part of the paper is a bit difficult to follow.

**Questions:**

* Can this approach work for other vision models, not only just the fully connected ones?

---

> ### Author Response · Authors · 2024-11-18
> **Response to Reviewer tMBJ**
>
> Thank you for your efforts to provide valuable comments on our work. We appreciate the reviewer's comments on what we need to consider more from an experimental point of view. We will add the part of the experiment as soon as possible.
> - - -
> **Response to Q1:**
>
> Are we correct in understanding that your question is asking whether our method can be applied not only to models with fully connected layers but also to models like CNNs?
>
> Unfortunately, our method is currently only applicable to fully connected layers.
>
> As of now, the DFA method does not perform well with convolution layers. According to *Crafton, Brian, et al. in "Direct feedback alignment with sparse connections for local learning" (Frontiers in Neuroscience 13, 2019)*, when DFA is applied to convolution layers, spatial information is not effectively learned, which hinders proper feature extraction. As a result, when fine-tuning a CNN model, the accuracy on ImageNet is 6.2%, while transferring learning with only fully connected layers yields an accuracy of 65.3%.
>
> We believe this may be because DFA does not fully capture the characteristic behavior of CNNs, where different features are extracted at each depth of the model while the same error is transmitted.
>
> Additionally, in the case of convolution layers with DFA, the feedback matrix transmits the error by unfolding the weights with dimensions (kernel_size, kernel_size, in_dim, out_dim) into a 2D form. Another challenge is that, in the feedback-weight matching method, one of the conditions for matching $W_l = F_{l-1} F_l$ is that $F_l^\top F_l = I$. Finding a feedback matrix that satisfies both $W_l = F_{l-1} F_l$ and $F_l^\top F_l = I$ for the 2D matrix $W_l$ is not straightforward.
>
> However, by developing additional methods to address these challenges, we hope to extend our approach to other vision models in the future.
> - - -
> **Response to W1:**
>
> It seems that we missed it because other papers rarely trained DFA in transformers. Thank you for your opinion. Our method will be effective in vision transformers with fully connected layers. We will do additional experiments and reply quickly.
> - - -
> **Response to W2:**
>
> The reason we did not use a large dataset is that our method does not utilize convolution layers. As mentioned earlier, since fully connected layers are used in DFA, the performance on large datasets like ImageNet tends to be low, making accurate evaluation difficult. This is why other DFA-related papers also rarely evaluate their methods on ImageNet.
>
> For this reason, we chose not to include results on large datasets. However, if our approach proves effective in vision transformers, it could certainly be extended to larger datasets in the future
> - - -
> **Response to W3:**
> We also recognize that it might be difficult to understand. We will try to make it a little easier to follow. If you could point out which parts are unclear, we will take that into account and revise accordingly.

---

> ### Author Response · Authors · 2024-11-22
> **Additional Response to Reviewer tMBJ**
>
> We would like to share the experimental results conducted on the ViT model. This experiment provided many valuable insights, and we appreciate your feedback and suggestions.
>
>
> The table below shows the results of training on the CIFAR-10, CIFAR-100, STL-10, and Imagenette datasets using ImageNet-21k pre-trained weights. All images were resized to 224x224 resolution. The values to the left and right of each '/' represent the accuracy at 5 epochs and 10 epochs, respectively, using the ViT model. "DFA ours" refers to feedback-weight matching, while "DFA fine" refers to fine-tuning from back-propagation weights.
>
> **Table 1 : Performance of ViT models**
> |   Model   |    Method   |        CIFAR-10       |         STL-10        |       Imagenette      |
> |:---------|:-----------|:---------------------:|:---------------------:|:---------------------:|
> | ViT-Small | DFA scratch |     0.322 / 0.428     |     0.221 / 0.243     |     0.230 / 0.228     |
> |           | DFA fine    |   0.378 / **0.475**   |     0.111 / 0.181     |     0.210 / 0.222     |
> |           | DFA ours    |   **0.392** / 0.435   | **0.247** / **0.259** | **0.319** / **0.386** |
> |  ViT-Tiny | DFA scratch |     0.281 / 0.384     |     0.197 / 0.234     |     0.168 / 0.230     |
> |           | DFA fine    |     0.332 / 0.388     |     0.164 / 0.226     |     0.209 / 0.238     |
> |           | DFA ours    | **0.397** / **0.429** | **0.247** / **0.254** | **0.294** / **0.309** |
>
> _Note: Bold indicates better result._
>
>
> In the ViT model, our method outperformed not only DFA from scratch but also fine-tuned DFA from back-propagation weights. This demonstrates that our approach is effective in the Vision Transformer model as well. However, it still underperforms compared to back-propagation. As the DFA results indicate, DFA currently appears inadequate for application to Vision Transformers.
>
>
> We hypothesize that delivering the same error across layers may impede the extraction of features with varying spatial characteristics, as seen in models like CNNs. This could explain the observed limitations. However, this assumption remains speculative due to a lack of supporting studies, highlighting the need for further research.
>
>
> We will consider addressing these challenges in CNN and vision transformer models as one of our key objectives in future research.

---

> ### Comment · Area_Chair_iXBs · 2024-11-25
>
> Dear Reviewer tMBJ,
>
> Could you kindly review the rebuttal thoroughly and let us know whether the authors have adequately addressed the issues raised or if you have any further questions.
>
> Best,
>
> AC of Submission6057

---

### Official Review · Reviewer_R6RF · 2024-11-07

**Soundness:** 2
**Presentation:** 2
**Contribution:** 3
**Rating:** 5
**Confidence:** 2

**Summary:**

In this work, the authors analyzed the direct feedback alignment (DFA) for fine-tuning network pertained via backpropagation and identified its limitations compared. Accordingly, the feedback-weight matching is proposed to enhance the ability and stability of DFA in fine-tuning and combined it with weight decay to further reduce the network output error. Experiments are conducted on various datasets.

**Strengths:**

1. *Originality*: It is novel to recognize and scrutinize that the strong weight alignment condition for DFA in the case of fine-tuning is unlikely to satisfy. In order to address this issue, the feedback-weight matching and weight decay for it are originally proposed to enhance the fine-tuning performance.

2. *Significance*: Fine-tuning non-fully connected networks has been an actively studied topic in the recent years, especially after the debut of ChatGPT. This work is of significance to analyze DFA for fine-tuning and propose the solutions: feedback-weight matching.

**Weaknesses:**

1. *Quality*: It would largely improve the quality of this work to articulate the proof of propositions in mathematically standard manners. Usually propositions are presented in the main content and their mathematical proofs are rigorously detailed in appendices.

2. *Clarity*: Please also have a careful double check on the boldface numbers in Tables 1 and 2. It looks confusing and misleading to highlight the performance of the proposed method instead of the best performance. The experimental results should also be further analyzed. Please see **Questions** for details.

**Questions:**

1. Would the authors clarify the statement "Equation (3) is referred to as weak weight alignment (WA) (Refinetti et al.,
2021), representing the state where no particular relationship exists between $\mathbf{W}_{1< l< L}^t$ and $\mathbf{F}_{l}\mathbf{A}_{l}^{t} \mathbf{F}_{l-1}^{\top}$, and between $\mathbf{W}_{L}^t$ and $\mathbf{A}_{L}^t \mathbf{F}_{L-1}^\top$? It seems that from Equation (3), $\mathbf{W}_{1< l< L}^t = \mathbf{F}_{l}\mathbf{A}_{l}^t\mathbf{F}_{l-1}^\top$, and $\mathbf{W}_{L}^{t}$ and $\mathbf{W}_{1< l< L}^t = \mathbf{A}_{L}^{t} \mathbf{F}_{L-1}^{\top}$.

2. In Table 2, the performance of DFA for training from scratch is the best among all three compared methods. It seems to be inconsistent to the performance on CIFAR-100. Could the authors explain the possible reasons?

3. Why is the dataset STL more difficult to fine-tune on?

---

> ### Author Response · Authors · 2024-11-18
> **Response to Reviewer R6RF**
>
> Thank you for taking the time to review our paper. We greatly appreciate your thoughtful and insightful feedback, which has been invaluable to us. We will revise the paper based on your suggestions. Below, we have provided detailed answers to your questions.
> - - -
> **Response to Q1:**
>
> We understand your opinion, and we apologize for presenting this in a way that makes it difficult to understand.
>
> The expression we used, “between $W_{1<l<L}^t$ and $F_{l-1}^\top$,” is represented as an equality in Equation (3), but our text states that there is no relationship, which seems to have made it even more difficult to understand.
>
> What we intended to explain is that in weak alignment, the relationship between $W_l$ and $F_l$ cannot be determined. This is because, in weak alignment, the alignment matrix term $A_l$prevents a clear definition of the relationship between $W_l$ and $F_l$. The alignment matrix contains a combination of inputs and errors, which makes it challenging to establish a clear connection.
>
> We will revise the original text by removing the $A_l$ term from the explanation and explicitly state that no relationship can be identified between $W_l$ and $F_l$. Thank you for pointing out this overlooked issue.
>
> “Equation (3) is referred to as weak weight alignment (WA) (Refinetti et al., 2021), representing the state where no particular relationship exists between $W_{1<l<L}^t$ and $F_lF_{l-1}^\top$ and between $W_L^t$ and $F_{L-1}^\top$”
> - - -
>
> **Response to Q2 and W2:**
>
> We understand that you interpreted the results to mean that among the DFA methods, DFA from scratch achieves the highest performance, and in Table 1, using CIFAR-100, our method achieves the highest performance, which seems contradictory and inconsistent. Is our understanding correct?
>
> To address this, it seems there might have been a slight misunderstanding in interpreting the data.
>
> While there are cases where DFA from scratch achieves high performance, in most cases—except for training BERT-Small on the MNLI and RTE datasets—our method outperforms DFA from scratch. So, we believe that both tables show consistent results that demonstrate the superiority of our method.
>
> We feel that this misunderstanding could be related to the lack of detail in the table description.
>
> We will add a statement to the table description indicating that bold values represent the high-performance results among the fine-tuning methods using DFA.
> - - -
> **Response to Q3:**
>
> We also recognize that dataset STL has lower performance. We believe that it is due to the following reasons.
>
> - The STL dataset has 10 times smaller training data than CIFAR-10. STL10 has only 500 samples per class, and CIFAR100 has 5,000 samples per class.
> - We resized the original 96x96 images to 32x32 for training. The reason for resizing to 32x32 was to evaluate fine-tuning performance using the same pre-trained weights; changing the input size would alter the weight dimensions of the fully connected layer. We think it affected the lower result because we used data with smaller size than the original data.
> - - -
> **Response to W2:**
>
> The proof is short, and we intended for it to be understood while reading the paper, but it seems to have caused confusion. To make it clearer, we will move the proof to the appendix and revise the paper.
>
> Thank you for your feedback. It will certainly help make the paper better.

---

> > ### Comment · Reviewer_R6RF · 2024-11-26
> >
> > I thank the authors for their reply. Although most of my questions are answered, the proofs are not updated accordingly. Hence I would keep my score.

---

> > > ### Author Response · Authors · 2024-11-26
> > > **Response to Reviewer R6RF**
> > >
> > > Thank you for your response. We have uploaded the revised version of the paper with the following updates:
> > >
> > > ---
> > > **Revised Sentence Expression:**
> > >
> > > The sentence has been updated to: "Equation (3) is referred to as weak weight alignment (WA) (Refinetti et al., 2021), representing the state where no particular relationship exists between $W_{1<l<L}^t$ and $F_lF_{l-1}^\top$ and between $W_L^t$ and $F_{L-1}^\top$."
> > >
> > > **Added Explanation for the Table:**
> > >
> > > An explanation of the boldface notation has been added to the table.
> > >
> > > **Moved Proof to Appendix:**
> > >
> > > The proofs have been relocated to Appendix A, and the corresponding equations for the proofs are included in the main text.
> > >
> > > ---
> > > We would be truly grateful if you could kindly take a moment to review this.

---

> ### Comment · Area_Chair_iXBs · 2024-11-25
>
> Dear Reviewer R6RF,
>
> Could you kindly review the rebuttal thoroughly and let us know whether the authors have adequately addressed the issues raised or if you have any further questions.
>
> Best,
>
> AC of Submission6057

---

### Author Response · Authors · 2024-11-28
**General Response**

We sincerely appreciate the reviewer's effort and constructive feedback, which have significantly contributed to improving our paper.

Below, we summarize the revisions we have made based on the reviewer's suggestions:

---

1. The proofs section, originally in the main manuscript, has been moved to Appendix A. We have also updated the relevant equations (Eq. 4 and Eq. 5) in the manuscript to reflect this change.

2. As per the reviewer's suggestion, we have revised the captions for Tables 1 and 2 to provide clearer descriptions.

3. We have revised the manuscript to improve clarity, particularly in lines 165-166, where we have refined the text to enhance readability and coherence.

---

We have made every effort to address the feedback and improve the quality of the manuscript. We kindly request the reviewers to let us know if any uncertainties remain, so we can address them before the discussion period concludes. Thank you.

---

### Meta-Review · Area_Chair_iXBs · 2024-12-19

**Metareview:**

(a) The paper introduces feedback-weight matching, a strategy that combines Direct Feedback Alignment (DFA) and backpropagation to enhance fine-tuning of pretrained networks, demonstrating improved performance over DFA but not surpassing backpropagation-based methods.

(b) Strengths: The paper demonstrates originality by identifying the limitations of strong weight alignment in DFA during fine-tuning and proposing feedback-weight matching with weight decay as a novel solution. The significance lies in analyzing DFA for fine-tuning rather than training from scratch, which is particularly relevant given the growing focus on fine-tuning in AI, especially with models like ChatGPT. The approach improves stability and performance through weight decay and feedback-weight matching and is validated on challenging transformer models for NLP tasks, showcasing its broader applicability.

(c) Weaknesses: The paper would benefit from clearer articulation of mathematical proofs, with rigorous details provided in appendices to enhance quality. Clarity issues include misleading boldface highlights in Tables 1 and 2 and inconsistent results, such as DFA's superior performance in one task but not others, which require further analysis. The evaluation is limited to a small number of datasets and lacks testing on widely used transformer models like ViT or Imagenet-trained models, raising questions about generalizability. While the proposed method improves upon DFA, its performance remains inferior to back-propagation-based approaches, and the improvements may largely result from incorporating back-propagation optimizations. Some parts of the paper are difficult to follow, impacting overall readability.

(d) The most important reasons for reject are: The paper lacks a clear explanation of the significant technical distinctions between the proposed DFA with Weight Decay and existing methods, which constitutes its core contribution. Additionally, the superior performance of DFA in fine-tuning may largely stem from the pretrained model rather than the proposed strategy, raising questions about the true source of the improvement. Further, the applying DFA to finetuning consistently underperforms relative to back-propagation and is limited to a narrower range of architectures. We understand that, rather than aiming to surpass back-propagation, the authors focus on improving the performance of DFA. However, given that the aforementioned limitation is highly evident, what practical value or significance does it hold?

**Additional Comments On Reviewer Discussion:**

(a) Reviewer R6RF shows that the quality of the work could be improved by presenting the propositions in a mathematically rigorous and standard manner, with detailed proofs included in the appendices. They note that the boldface numbers in Tables 1 and 2 are misleading, as they highlight the proposed method's performance instead of the best results. Additionally, the reviewer suggests further analysis of the experimental results to enhance clarity and understanding. The authors address most of the concerns, but fail to update the proofs promptly. Therefore the reviewer decides to keep the score of 5.

(b) Reviewer tMBJ finds that the proposed approach has not been tested on other transformer models, such as ViT, which raises questions about its generalizability. The method has been evaluated on a limited set of datasets, and further testing on models trained or fine-tuned on ImageNet is suggested. Additionally, some parts of the paper are difficult to follow and could benefit from clarification. The authors seem to address most of the concerns, the reviewer fails to have a response even after the AC has sent the remind.

(c) Reviewer 49ZY shows that the suggested approach is ineffective, yielding inferior results compared to back-propagation methods. While the method aligns Direct Feedback weights with back-propagation weights, the improvement is likely due to back-propagation optimization. The reviewer requests clarification on whether the enhanced performance truly stems from the proposed approach or from back-propagation. The initial rebuttal has resolved most doubts of the Reviewer 49ZY, but he/she further raises more concerns regarding   weight decay, derivation and exaggeration.

---

### Decision · Program_Chairs · 2025-01-22

Reject